# Disruptions in Lung Cancer Detection During COVID-19

**DOI:** 10.3390/cancers16234001

**Published:** 2024-11-29

**Authors:** Trisha Lal, Uriel Kim, Christina S. Boutros, Natalie N. Chakraborty, Susan J. Doh, Christopher W. Towe, Richard S. Hoehn

**Affiliations:** 1Division of Surgical Oncology, University Hospitals Cleveland Medical Center, Cleveland, OH 44106, USA; trisha.lal@uhhospitals.org (T.L.); christina.boutros@uhhospitals.org (C.S.B.); natalie.chakraborty@uhhospitals.org (N.N.C.); susan.doh@uhhospitals.org (S.J.D.); 2School of Medicine, Case Western Reserve University, Cleveland, OH 44106, USA; uxk13@case.edu (U.K.); christopher.towe@uhhospitals.org (C.W.T.); 3Department of Population and Quantitative Health Sciences, School of Medicine, Case Western Reserve University, Cleveland, OH 44106, USA; 4Case Comprehensive Cancer Center, School of Medicine, Case Western Reserve University, Cleveland, OH 44106, USA; 5Department of Internal Medicine, Cedars-Sinai Medical Center, Los Angeles, CA 90048, USA; 6Division of Thoracic Surgery, University Hospitals Cleveland Medical Center, Cleveland, OH 44106, USA

**Keywords:** diagnosing lung cancer, disparities, high-risk populations, non-small-cell lung cancer, COVID-19

## Abstract

The COVID-19 pandemic disrupted timely diagnosis of many cancers, including lung cancer, the leading cause of cancer-related death. In this study, we analyzed national data to understand how lung cancer detection was affected during the first two years of the pandemic, focusing on differences across demographic groups and communities. We found that lung cancer detection dropped significantly in 2020, particularly for rural residents, non-Hispanic Black and Asian individuals, and women. Although detection rates started to recover in 2021, disparities persisted, particularly in rural areas. Our findings highlight a hidden pool of undiagnosed cases and the need for targeted interventions to reach underserved populations. These insights could help guide future public health strategies to reduce disparities and improve early lung cancer detection for all groups.

## 1. Introduction

Lung cancer is the leading cause of cancer-related death worldwide [1], with advanced-stage disease posing a significant mortality burden. Nearly half of lung cancer cases are diagnosed only after distant spread, at which point 5-year survival is less than 10% [2]. Accordingly, early detection of lung cancer, which includes prompt diagnosis following onset of symptoms or incidental findings during unrelated imaging, is crucial to improving outcomes [3]. Despite advancements in imaging and diagnostic technologies, substantial disparities in detection and outcomes persist [4]. Historically marginalized populations, including Black patients and those who reside in rural communities, continue to face higher rates of late-stage diagnosis and delays in guideline-concordant treatment [5,6]. Additionally, financial barriers and socioeconomic inequalities contribute significantly to disparities in lung cancer detection, diagnosis, and adherence to follow-up care, highlighting the need for equitable access globally [5].

The COVID-19 pandemic led to widespread disruptions in the delivery of cancer care, further exacerbating inequities in lung cancer detection and diagnosis [7]. Marginalized populations were disproportionately affected due to pre-existing barriers to healthcare access that were intensified by the pandemic [8] and compounded by nationwide policies to limit COVID-19 exposure. For example, in New York City, the epicenter of the pandemic in the United States, all elective healthcare was halted from March to June 2020, resulting in a sharp decline in early-stage lung cancer diagnoses. Conversely, there was an increase in late-stage lung cancer diagnoses during pandemic surges, suggesting that primarily sicker and more symptomatic patients sought medical care during this period [9]. Numerous retrospective studies have observed this pattern of reduced early-stage lung cancer diagnoses and a corresponding increase in late-stage diagnoses during the pandemic, indicating a shift toward more advanced disease presentations as a consequence of reduced early detection [10]. Although these studies have effectively captured the immediate effects of the pandemic on lung cancer detection, they are limited in scope, often focusing on localized populations or short-term outcomes. There is a lack of comprehensive national data assessing the duration and extent of recovery in lung cancer detection and care, particularly for patients who missed early diagnoses during the pandemic’s peak.

Assessing the long-term impact of the COVID-19 pandemic on cancer detection and diagnosis is critical for shaping strategies that mitigate future disruptions and addressing entrenched health disparities. This study aims to quantify both the extent of disruption and the trajectory of recovery in lung cancer diagnoses during the first two years of the pandemic. We hypothesize that vulnerable populations will experience prolonged disparities in detection and recovery, reflecting deep-rooted inequities in healthcare access. By analyzing these trends, our findings will provide crucial insights to guide public health efforts in improving early lung cancer detection and reducing disparities in marginalized communities.

## 2. Materials and Methods

### 2.1. Data Source

We conducted an epidemiological analysis of lung cancer incidence from the April 2024 release of the Surveillance, Epidemiology, and End Results (SEER) database accessed using the SEER*Stat software Version 8.4.4. This dataset includes nationally representative cancer registry data through the end of 2021. SEER is a population-based database maintained by the National Cancer Institute (NCI), and it comprises data from various state registries that track cancer incidence and survival. Our study used the SEER-22 database, which includes cancer incidence data from 22 state registries, representing approximately 47.9% of the United States population [11]. This extensive coverage provides insights into population-level cancer trends across diverse demographic groups. 

### 2.2. Patient Cohort

Lung cancer cases from 2001 to 2021 were identified to estimate expected incidence trends; however, only cases diagnosed between 2018 and 2021 were included in the main analysis. The inclusion criteria included the following: (I) ICD-O-3 histologic code as lung and bronchus; (II) microscopically confirmed disease; (III) malignant; (IV) classified as lung primary by the American Joint Committee on Cancer (AJCC); and (V) age 20 to 85+. Cases were histologically classified as non-small-cell lung cancer (NSCLC), small-cell lung cancer (SCLC), or Other based on the SEER Program Coding and Staging Manual [12]. The exclusion criteria were as follows: (I) diagnosis between 2000 and 2017; (II) unknown age; and (III) more than one primary cancer or not the first of two or more primaries.

Data selection involved filtering the dataset using the inclusion and exclusion criteria to ensure that only relevant lung cancer cases were analyzed. Missing data, such as unknown age, were handled by excluding these cases to maintain the accuracy of the results. ICD-O-3 codes ensured that we consistently identified lung cancer cases across the SEER database, which does not always explicitly label cases as “lung cancer” without appropriate coding.

### 2.3. Exposure

Patients were grouped by year of diagnosis: pre-COVID-19 (2018–2019), year 1 of COVID-19 (2020), and year 2 of COVID-19 (2021).

### 2.4. Outcomes

The primary endpoint of this study is the lung cancer stage at diagnosis across three distinct periods: pre-COVID-19, year 1 of COVID-19 (2020), and year 2 of COVID-19 (2021). Secondary endpoints assessed temporal trends in lung cancer detection rates by stage across these periods and quantified the extent of recovery in detection rates between years 1 and 2 of the pandemic.

### 2.5. Statistical Analysis

Demographic and county-level characteristics were assessed using StataSE v16.1 (Statacorp LLC, College Station, TX, USA) software. We then utilized a previously described method [13,14] to compare the expected to the observed lung cancer incidence ratios during the first (2020) and second years (2021) of the COVID-19 pandemic. Expected lung cancer incidence was estimated by extrapolating trends in diagnosis between 2000 and 2019 to model the expected incidence in 2020 and 2021 using the NCI’s Joinpoint Regression Program. This method allows for estimating expected incidence rates by identifying points in time where statistically significant changes in trend occur and using those points to project future rates. The percent difference (PD) between the expected and observed incidences of lung cancer in 2020 and 2021 was then calculated to quantify the disruption and subsequent recovery of lung cancer detection during the pandemic. Stratified analyses were also performed based on demographic and county-level characteristics. Incidence rates were age-adjusted and delay-adjusted when appropriate. Delay adjustments were used as they represent projections to more closely approximate the rates after many submissions [15].

To understand changes in the risk of metastatic lung cancer diagnosis during the pandemic, we first conducted a multivariable logistic regression analysis to model the baseline odds of being diagnosed with metastatic disease based on a patient’s histologic, demographic (age, sex, race/ethnicity), and community (rurality, income level) characteristics during the pre-COVID-19 period (2018–2019). To evaluate how this risk evolved during the pandemic, we used propensity-score-adjusted analyses to estimate the percentage of patients presenting with metastatic disease at each time point, adjusting for changes in patient characteristics.

The analysis was performed using R Studio statistical software (version 2024.4.2.764). Statistical significance was determined using an alpha of 0.05.

### 2.6. Institutional Assurances

This project was exempt from our institution’s Institutional Review Board approval due to the dataset’s de-identified nature.

### 2.7. Declaration of Generative AI and AI-Assisted Technologies in the Writing Process

During the preparation of this work, the authors used ChatGPT and Grammarly to review grammar and maintain brevity. After using these services as needed, the authors reviewed and edited the content and take full responsibility for it.

## 3. Results

### 3.1. Study Population and Cohort Characteristics

A total of 259,495 patients with lung cancer diagnosed between 2001 and 2021 were included in the overall trend analysis. After applying exclusion criteria, 98,616 patients diagnosed with lung cancer between 2018 and 2021 were included in the analyses to focus specifically on the impact of the COVID-19 pandemic (Figure 1). The cohort was divided into three groups based on year of diagnosis: pre-COVID-19 (2018–2019, *n* = 51,883), year 1 of COVID-19 (2020, *n* = 22,472), and year 2 of COVID-19 (2021, *n* = 24,261). Demographic characteristics are summarized in Table 1. Overall, the study sample primarily comprised individuals who were male, older (60 years or older at diagnosis), non-Hispanic White, and living in large metropolitan counties.

This cohort diagram outlines the selection process for lung cancer patients included in the study from the SEER database. Of 259,495 identified cases between 2001 and 2021, 98,616 met inclusion criteria and were not excluded from analyses explicitly focused on the impact of the COVID-19 pandemic. The final cohort was categorized into three groups: pre-COVID (2018–2019, *n* = 51,883), year 1 of COVID (2020, *n* = 22,472), and year 2 of COVID (2021, *n* = 24,261).

### 3.2. Changes in Lung Cancer Incidence During COVID-19: Year 1 (2020)

Lung cancer detection decreased in 2020. The overall incidence was 45.25 per 100,000 individuals compared to the expected 50.25 per 100,000, representing a percent difference of −10% (95% CI: −8.3 to −11.6). Incidence dropped across nearly all examined subgroups, with the largest declines observed in non-Hispanic Asian (−13.9, 95% CI: −15.4 to −12.5) and Black (−10.9, 95% CI: −9.9 to −11.9) populations, females (−11.5, 95% CI: −11.2 to −11.7), and residents of medium metropolitan areas (−11.4%, 95% CI: −10.4 to −12.4).

Notably, age greater than 80 was associated with increased detection of lung cancer in 2020 (9.8%, 95% CI: 11.0 to 8.7), whereas age between 40 and 64 and 65 and 79 was associated with decreased detection in 2020 (−8.1%, 95% CI: −6.1 to −10.1; −9.6%, 95% CI: −8.1 to −11.0) (Table 2).

### 3.3. Changes in Lung Cancer Incidence During COVID-19: Year 2 (2021)

By 2021, lung cancer detection remained lower than expected, although the decline was smaller. The overall observed incidence rate was 46.93 per 100,000, a 5.0% decrease from the expected rate (95% CI: −3.2 to −6.7). Specifically, the detection of localized disease showed significant recovery, shifting to a 2.9% increase in 2021. Non-Hispanic American Indian/Alaska Native ethnicity (16.8%, 95% CI: 5.9 to 27.8) and residence in small metropolitan counties (6.7%, 95% CI: 6.3 to 7.2) were also associated with increased detection of lung cancer in 2021.

Similar trends were seen in individuals over 80, who showed an even greater detection rate (18.1%, 95% CI: 19.3 to 16.8) in 2021. Female (−5.6%, 95% CI: −5.4 to −5.9) sex and non-Hispanic Asian (−3.7%, 95% CI: −5.2 to −2.3) and Black (−3.2%, 95% CI: −2.0 to −4.4) ethnicity were also associated with improved detection rates compared to 2020, although rates remained below pre-pandemic projections.

In contrast, residence in rural areas not adjacent to metropolitan counties was associated with even lower detection rates in the second year of the pandemic compared to the first year (Table 2).

### 3.4. Adjusted Multivariate and Propensity Score Analysis of Distant Disease at Presentation

The adjusted multivariate analysis demonstrated that at baseline, during the pre-COVID-19 period (2018–2019), patients with SCLC had increased odds of distant disease at presentation (OR 2.58, 95% CI: 2.45–2.73) compared to the reference group of NSCLC, as did male patients and patients from racial/ethnic minorities (Table 3).

The propensity score analysis identified several significant associations related to distant disease presentation during the COVID-19 pandemic. SCLC patients consistently had higher odds of distant disease presentation than NSCLC patients across all study periods, with consistently high adjusted percentages of distant disease presentation: 74.90% pre-COVID, 74.42% in 2020, and 74.80% in 2021.

Female patients consistently had lower rates of distant disease compared to males, with adjusted percentages of 52.27% pre-COVID, 54.32% in 2020, and 52.56% in 2021, while male patients maintained a rate of around 57% across all periods.

Non-Hispanic Asian/Pacific Islander patients had higher adjusted percentages of distant disease compared to non-Hispanic White patients throughout all periods, with rates of 65.93% in 2020 and 64.08% in 2021 versus 52–54% in NH White patients. Hispanic and non-Hispanic Black patients also showed persistently higher percentages of distant disease presentation through 2021. Individuals over 65 experienced lower adjusted percentages of distant disease than those aged 50–64 across all periods. The adjusted percentage for patients aged 65+ was 52.63% pre-COVID-19 and 52.9% in 2021, slightly increasing to 54.97% in 2020. Rurality and median income were not significantly associated with changes in distant disease presentation.

## 4. Discussion

Our study is notable for its use of nationally representative SEER data to comprehensively quantify the impact of the COVID-19 pandemic on lung cancer incidence and diagnosis. We found a 10% decline in lung cancer incidence during the first year of the pandemic (2020) compared to pre-pandemic trends. This reduction was most pronounced for localized and regional disease, with disproportionately higher impacts observed in females, non-Hispanic Black, non-Hispanic Asian, and rural populations. Importantly, by 2021, we documented partial recovery in lung cancer incidence, reflecting improvements in detection efforts, although overall rates remained 5% below expected levels. Our findings highlight that while localized disease diagnosis significantly rebounded, regional and distant disease diagnoses remained persistently lower, particularly in rural communities. In our adjusted analyses that examined the risk of being diagnosed with metastatic disease, patients with SCLC and those of Hispanic, non-Hispanic Black, or Asian/Pacific Islander ethnicity had persistently higher percentages of distant disease presentation through 2021. This study underscores the long-term disruptions in cancer care delivery, particularly among vulnerable populations, and emphasizes the need for targeted efforts to address these disparities.

Our findings of lower-than-expected lung cancer incidence during the pandemic are consistent with previous studies reporting declines in cancer diagnoses during this time [13,16,17]. Similar trends have been observed globally, such as in Italy and Canada, where disruptions in lung cancer diagnosis during the pandemic also led to decreases in new cases and shifts toward more advanced-stage disease [18,19]. These decreases in early-stage incidence, alongside increases in advanced-stage diagnoses in certain populations, reflect a likely stage migration, which is multifactorial in etiology. In 2020, the American College of Chest Physicians recommended delaying screening for new patients and postponing annual LDCT scans for patients already enrolled in lung cancer screening programs to reduce the risk of COVID-19 exposure [20]. Consequently, there was a marked reduction in patients receiving low-dose CT scans in the first year of the pandemic. Even as screening gradually resumed, new patient screenings remained low. Additionally, missed healthcare appointments and fewer new patient referrals contributed to a surge in advanced-stage cases [21,22,23,24]. As a result, the pandemic not only disrupted early diagnosis efforts, thus reducing the number of new diagnoses, but also set the stage for a rise in advanced-stage diagnoses, underscoring the urgent need to reimplement screening to prevent cancer-stage migration.

Our findings emphasize the importance of screening in improving lung cancer outcomes. The US National Lung Screening Trial (NSLT) demonstrated that early detection through low-dose CT (LDCT) screening can reduce lung cancer mortality by at least 20% among high-risk individuals compared to the prior standard of chest radiography [25]. The NELSON trial subsequently showed a reduction in lung-cancer-specific mortality of 24% in men and 33% in women who underwent LDCT screening compared to those who did not undergo screening [26]. These findings highlight the critical role of proactive screening in reducing mortality and suggest that disruptions in screening during the pandemic may have substantially contributed to the observed decline in early-stage lung cancer detection. Notably, a study conducted in the US demonstrated that the mortality-reduction benefit of lung cancer screening is more favorable in African American individuals than in white individuals, which underscores the need to ensure equitable access to screening for all populations [27]. Additionally, a study performed in Asia found that financial barriers and socioeconomic inequalities significantly contribute to disparities in lung cancer screening adherence, emphasizing the need to address these issues globally to ensure equitable access to early detection [28].

The propensity score analysis also highlighted significant disparities in distant disease presentation between histological subtypes of lung cancer. SCLC patients consistently had higher rates of distant disease presentation compared to NSCLC patients across all study periods. This finding aligns with the aggressive nature of SCLC, which is frequently asymptomatic until advanced stages [29]. The pandemic likely exacerbated these trends due to disruptions in healthcare access. Moreover, LDCT has shown limited benefit in SCLC, with most cases still being diagnosed at advanced stages with screening [30]. These limitations emphasize the need for innovative early detection strategies, particularly during public health crises, to improve outcomes for SCLC patients.

Our analysis also found that the increased likelihood of distant disease presentation was more prevalent among historically marginalized populations, including non-Hispanic Black patients and Asian/Pacific Islander patients. This pattern, which persisted from pre-pandemic years, has been consistently reported in prior studies [5,6], highlighting ongoing racial disparities in lung cancer outcomes during the pandemic. Similarly, non-Hispanic Asian/Pacific Islander patients were more likely to be diagnosed with distant disease both before and during the pandemic. Stage migration was also observed in individuals older than 65 and those in small metropolitan areas, who were more likely to present with advanced disease in 2020. However, by 2021, the likelihood of distant disease presentation had decreased in these groups, aligning with prior studies that reported an increased likelihood of distant disease at diagnosis during the early phases of the pandemic [9,31]. However, other studies have not found statistically significant stage migration [32,33], suggesting that our study’s ability to stratify patients into subgroups based on sociodemographic or county-level characteristics may reveal more nuanced disparities in the impact of the pandemic on lung cancer diagnosis.

This study has several limitations that should be acknowledged. First, while the SEER database provides extensive, nationally representative data, it only covers 22 state cancer registries, which may introduce selection bias. Additionally, our analysis was contingent on the assumption that pre-pandemic trends in lung cancer incidence would have continued unchanged, potentially overlooking other factors, such as shifts in smoking behavior or healthcare access, that could have influenced lung cancer detection and diagnosis during the pandemic. Another limitation is the absence of data on public attitudes toward lung cancer screening during the pandemic. Understanding perceptions and behaviors could provide valuable context for the observed trends and help inform future interventions. The disparities observed in recovery across different subgroups may be influenced by unmeasured confounders, such as variations in the perceived risk of COVID-19 exposure or hospital-specific delays in reinstating screening programs. Moreover, the retrospective design of this study precludes causality. Although we employed robust statistical methods to project the expected lung cancer incidence, these projections carry inherent uncertainties and may not fully capture the complex dynamics of the evolving healthcare landscape during this period.

Despite these limitations, our findings underscore the significant impact of the COVID-19 pandemic on lung cancer detection and diagnosis, particularly for vulnerable populations. The decline in early-stage lung cancer detection in 2020 and the persistent under-diagnosis of advanced-stage disease in 2021 suggest a hidden reservoir of undiagnosed cases that could contribute to future burdens of advanced disease. The recovery of localized disease detection in 2021 demonstrates that restarting LDCT screening programs and public health initiatives can effectively mitigate pandemic-related disruptions. Public health efforts must prioritize reengaging patients, particularly in rural and minoritized areas, to ensure that all populations benefit from screening recovery efforts. Persistently low detection rates for advanced-stage disease emphasize the need for targeted interventions to ensure timely diagnosis and equitable healthcare access for all populations, especially those disproportionately affected by pandemic-related disruptions.

## 5. Conclusions

In conclusion, the COVID-19 pandemic has had a profound and lasting impact on lung cancer detection and diagnosis in the United States, with a significant decline in new cases identified during the first year of the pandemic and only partial recovery by the second year. Our analysis reveals that these disruptions were most severe for early-stage disease and disproportionately affected vulnerable populations, including females, non-Hispanic Black and Asian individuals, and those residing in rural areas. While there has been some improvement in diagnosis rates, particularly for localized disease, the persistence of disparities in recovery underscores the urgent need for targeted public health interventions. To mitigate the long-term consequences of these disruptions, efforts must focus on reengaging underserved communities, expanding access to screening programs, and addressing the structural inequities exacerbated by the pandemic. These findings emphasize the importance of building resilient healthcare systems that adapt to crises while ensuring equitable care for all populations. Moving forward, examining lung cancer survival trends during this period will be crucial to fully understand the pandemic’s impact and to inform targeted interventions.

## Figures and Tables

**Figure 1 cancers-16-04001-f001:**
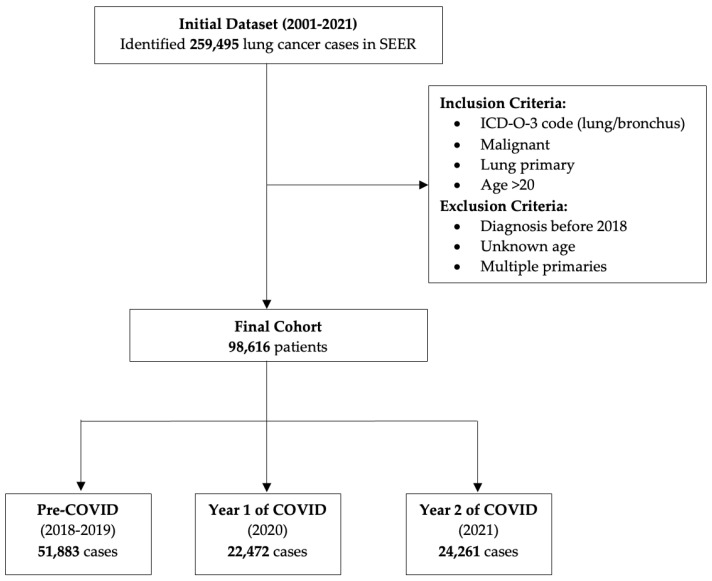
Flowchart of study cohort selection for lung cancer.

**Table 1 cancers-16-04001-t001:** Demographic and clinical characteristics of lung cancer patients, 2018–2021.

		Year of Diagnosis			*p*-Value
		2018–2019	2020	2021	
** *n* **		51,883	22,472	24,261	
**All Lung Cancers**	Localized	11,054 (21.3%)	4537 (20.2%)	5345 (22.0%)	<0.001
	Regional	10,860 (20.9%)	4513 (20.1%)	4862 (20.0%)	
	Distant	29,969 (57.8%)	13,422 (59.7%)	14,054 (57.9%)	
**Sex**	Male	26,783 (51.6%)	11,648 (51.8%)	12,293 (50.7%)	0.02
	Female	25,100 (48.4%)	10,824 (48.2%)	11,968 (49.3%)	
**Age Group**	20–49	1591 (3.1%)	662 (2.9%)	704 (2.9%)	<0.001
	50–59	7539 (14.5%)	3110 (13.8%)	3052 (12.6%)	
	60–69	17,459 (33.7%)	7734 (34.4%)	8391 (34.6%)	
	70–79	17,395 (33.5%)	7687 (34.2%)	8276 (34.1%)	
	≥80	7899 (15.2%)	3279 (14.6%)	3838 (15.8%)	
**Race/Ethnicity**	Hispanic (any race)	3776 (7.3%)	1666 (7.4%)	1834 (7.6%)	0.003
	Asian	4548 (8.8%)	2044 (9.1%)	2251 (9.3%)	
	NH Black	5940 (11.4%)	2448 (10.9%)	2754 (11.4%)	
	NH others and unknowns	470 (0.9%)	239 (1.1%)	273 (1.1%)	
	NH White	37,149 (71.6%)	16,075 (71.5%)	17,149 (70.7%)	
**Rurality**	Large metropolitan	27,065 (52.2%)	11,628 (51.7%)	12,583 (51.9%)	0.53
	Medium metropolitan	11,301 (21.8%)	4897 (21.8%)	5312 (21.9%)	
	Small metropolitan	4798 (9.2%)	2096 (9.3%)	2242 (9.2%)	
	Rural, adjacent to a metropolitan area	4969 (9.6%)	2120 (9.4%)	2295 (9.5%)	
	Rural, not adjacent to a metropolitan area	3664 (7.1%)	1689 (7.5%)	1775 (7.3%)	
**Income**	<USD 50,000	4861 (9.4%)	1830 (8.1%)	1956 (8.1%)	<0.001
	USD 50,000-USD 64,999	9337 (18.0%)	3946 (17.6%)	4234 (17.5%)	
	USD 65,000-USD 79,999	10,528 (20.3%)	4492 (20.0%)	4858 (20.0%)	
	USD 80,000-USD 94,999	14,437 (27.8%)	6171 (27.5%)	6761 (27.9%)	
	≥USD 95,000	12,720 (24.5%)	6033 (26.8%)	6452 (26.6%)	

This table summarizes the demographic and clinical characteristics of lung cancer patients diagnosed during the pre-COVID-19 period (2018–2019), year 1 of COVID-19 (2020), and year 2 of COVID-19 (2021). Variables include sex, age group, race/ethnicity, rurality, and income. Statistical significance was defined as *p* < 0.05 and calculated to compare distributions between the three periods. NH: Non-Hispanic.

**Table 2 cancers-16-04001-t002:** Observed and expected lung cancer incidence rates during the COVID-19 pandemic.

	Pandemic Year 1 (2020)			Pandemic Year 2 (2021)		
	*Incidence per 100,000*			*Incidence per 100,000*		
	Expected	Observed	Percent Difference (95% CI)	Expected	Observed	Percent Difference (95% CI)
**All Lung Cancers**	50.25	45.25	**−10.0** (−8.3 to −11.6)	49.39	46.93	**−5.0** (−3.2 to −6.7)
Localized	13.19	12.17	**−7.8** (−3.9 to −11.6)	12.82	13.19	**2.9** (7.2 to −1.4)
Regional	9.66	8.62	**−10.8** (−5.7 to −15.9)	9.26	8.94	**−3.5** (2.0 to −9.0)
Distant	21.41	20.88	−2.5 (3.4 to −8.4)	20.50	20.80	1.5 (7.6 to −4.7)
**Sex**						
Female	46.48	41.14	**−11.5** (−11.2 to −11.7)	46.05	43.45	**−5.6** (−5.4 to −5.9)
Male	55.54	50.84	**−8.5** (−6.0 to −10.9)	54.14	51.77	**−4.4** (−1.8 to −7.0)
**Race/Ethnicity**						
Hispanic (any race)	27.07	23.60	−12.8 (−13.7 to −12.0)	26.69	25.53	−4.4 (5.2 to −3.5)
NH AI/AN	49.51	49.28	−0.5 (−10.9 to 9.9)	49.18	57.46	**16.8** (5.9 to 27.8)
NH Asian/PI	35.05	30.16	**−13.9** (−15.4 to −12.5)	34.77	33.47	**−3.7** (−5.2 to −2.3)
NH Black	53.55	47.71	**−10.9** (−9.9 to −11.9)	52.41	50.73	**−3.2** (−2.0 to −4.4)
NH White	53.98	51.63	−4.4 (1.9 to −9.7)	51.97	52.98	1.9 (7.6 to −3.7)
**Age Group**						
<20	0.09	0.07	−16.3 (−49.0 to 16.5)	0.09	0.09	−2.0 (−38.1 to 34.1)
20–39	1.12	1.03	−7.5 (−14.7 to −0.2)	1.09	1.01	−7.3 (−14.8 to 0.1)
40–64	39.83	36.61	**−8.1** (−6.1 to −10.1)	38.90	37.16	**−4.5** (−2.4 to −6.6)
65–79	279.98	253.18	**−9.6** (−8.1 to −11.0)	274.82	263.42	**−4.1** (−2.6 to −5.7)
80+	279.98	274.82	**9.8** (11.0 to 8.7)	274.82	324.45	**18.1** (19.3 to 16.8)
**County Characteristics**						
*Rurality*						
Large metropolitan	43.30	41.33	−4.6 (2.7 to −11.8)	41.17	41.93	1.9 (9.6 to −5.8)
Medium metropolitan	50.61	44.85	**−11.4** (−10.4 to −12.4)	49.71	47.05	**−5.3** (−4.2 to −6.5)
Small metropolitan	50.61	52.62	**4.0** (4.4 to 3.5)	49.71	53.05	**6.7** (7.2 to 6.3)
Rural, adjacent to metropolitan area	65.36	59.68	**−8.7** (−8.2 to −9.2)	64.67	59.97	**−7.3** (−6.8 to −7.8)
Rural, not adjacent to metropolitan area	64.08	58.86	**−8.1** (−8.8 to −7.5)	63.32	58.09	**−8.3** (−8.9 to −7.6)
*Poverty*						
<10%	46.64	42.34	**−9.2** (−7.4 to −11.1)	45.66	42.54	**−6.8** (−4.9 to −8.7)
10–19.99%	51.16	45.65	**−10.8** (−9.5 to −12.1)	50.30	46.62	**−7.3** (−6.0 to −8.7)
20%+	55.35	49.23	**−11.1** (−10.7 to −11.4)	54.34	52.07	**−4.2** (−3.8 to −4.6)
*% Foreign-Born*						
<10%	63.26	57.31	**−9.4** (−8.5 to −10.3)	62.42	58.06	**−7.0** (−6.1 to −7.9)
10–19.99%	49.99	44.27	**−11.4** (−10.7 to −12.2)	49.12	44.86	**−8.7** (−7.9 to −9.4)
20%+	40.29	35.93	**−10.8** (−8.8 to −12.9)	39.34	36.87	**−6.3** (−4.1 to −8.5)
*% No High School Education*						
<10%	53.19	47.82	**−10.1** (−8.7 to −11.5)	52.27	48.95	**−6.4** (−4.8 to −7.9)
10–19.99%	48.07	43.04	**−10.5** (−8.9 to −12.1)	47.17	43.58	**−7.6** (−5.9 to −9.3)
20%+	45.23	40.70	**−10.0** (−9.2 to −10.8)	44.14	41.92	**−5.0** (−4.2 to −5.9)

This table presents the observed and expected lung cancer incidence rates for 2020 and 2021 compared to pre-pandemic projections (2018–2019). Percent differences between expected and observed incidence rates are shown across all lung cancer stages and key demographic groups. The table shows a 10% decline in lung cancer detection in 2020, disproportionately affecting females, non-Hispanic Black and Asian populations, and rural communities. By 2021, localized disease detection improved, but advanced-stage diagnoses remained below expected levels, particularly in rural areas. AI/AN: American Indian/Alaska Native. PI: Pacific Islander. Bolded figures indicate statistically significant (*p* < 0.05) values.

**Table 3 cancers-16-04001-t003:** Adjusted multivariate and propensity score analysis of distant disease at presentation.

Distant Disease	Adjusted Odds Ratio (95% CI, *p*-Value)	% of Patients with Distant Disease, Adjusted (95% CI)			
	Pre-COVID-19	Pre-COVID-19	2020	2021	*p*
**Histology**					
NSCLC	**Ref**	54.58% (54.11–55.04%)	56.8% (56.09–57.51%)	54.72% (54.03–55.40%)	<0.001
SCLC	2.58 (2.45–2.73, *p* < 0.001)	74.90% (73.96–75.85%)	74.42% (73.01–75.83%)	74.80% (73.43–76.17%)	0.84
Other	0.71 (0.68–0.73, *p* < 0.001)	45.42% (44.65–46.20%)	47.41% (46.31–48.51%)	45.15% (44.06–46.24%)	<0.01
**Sex**					
Female	**Ref**	52.27% (51.73–52.80%)	54.32% (53.52–55.13%)	52.56% (51.79–53.34%)	<0.001
Male	1.23 (1.19–1.27, *p* < 0.001)	57.45% (56.93–57.98%)	59.23% (58.46–60.01%)	57.28% (56.50–58.05%)	<0.001
**Race/Ethnicity**					
NH White	**Ref**	53.09% (52.65–53.54%)	54.70% (54.04–55.36%)	52.48% (51.84–53.13%)	<0.001
Hispanic	1.40 (1.32–1.49, *p* < 0.001)	60.49% (59.14–61.85%)	62.48% (60.47–64.50%)	60.73% (58.80–62.67%)	0.25
NH AI/AN	1.39 (1.10–1.77, *p* < 0.001)	62.01% (56.59–67.44%)	60.15% (51.93–68.37%)	61.45% (53.47–69.43%)	0.93
NH Asian/PI	1.49 (1.41–1.58, *p* < 0.001)	61.65% (60.39–62.91%)	65.93% (64.11–67.75%)	64.08% (62.33–65.83%)	<0.01
NH Black	1.19 (1.13–1.25, *p* < 0.001)	57.44% (56.32–58.55%)	59.43% (57.74–61.13%)	59.77% (58.13–61.41%)	0.072
**Age Group**					
50–64	**Ref**	59.61% (58.96–60.26%)	60.55% (59.58–61.53%)	59.27% (58.29–60.24%)	0.114
65+	0.78 (0.76–0.81, *p* < 0.001)	52.63% (52.17–53.09%)	54.97% (54.29–55.65%)	52.85% (52.20–53.51%)	<0.001
**County Characteristic**					
*Rurality*					
Large metropolitan	**Ref**	54.95% (54.43–55.47%)	56.94% (56.17–57.71%)	55.77% (55.02–56.52%)	<0.001
Medium metropolitan	1.01 (0.97–1.05, *p* = 0.66)	55.17% (54.37–55.98%)	56.25% (55.05–57.45%)	53.15% (50.56–55.74%)	0.056
Small metropolitan	0.95 (0.90–1.01, *p* = 0.08)	53.73% (52.48–54.97%)	57.86% (56.02–59.69%)	52.74% (50.93–54.56%)	<0.001
Rural, adjacent to metropolitan area	0.99 (0.94–1.06, *p* = 0.86)	54.76% (53.54–55.99%)	57.18% (55.35–59.02%)	54.09% (52.30–55.89%)	0.025
Rural, not adjacent to metropolitan area	1.01 (0.94–1.08, *p* = 0.88)	55.33% (53.92–56.75%)	55.95% (53.89–58.01%)	54.72% (52.68–56.75%)	0.647
*Median Income*					
High income	**Ref**	54.99% (53.90–56.09%)	57.05% (55.40–58.71%)	52.66% (48.01–57.31%)	0.135
Middle income	0.99 (0.94–1.04, *p* = 0.58)	54.77% (54.34–55.19%)	56.67% (56.05–57.30%)	54.97% (54.36–55.57%)	<0.001
Low income	1.00 (0.92–1.08, *p* = 0.91)	55.91% (54.67–57.15%)	57.77% (55.78–59.75%)	54.64% (52.68–56.59%)	0.052

This table presents adjusted odds ratios (ORs), 95% confidence intervals (CIs), and adjusted percentages for distant disease presentation among lung cancer patients. The multivariate regression analysis, conducted for the pre-COVID-19 period (2018–2019), modeled the independent odds of being diagnosed with distant disease to characterize baseline risks based on patient histologic, demographic, and community characteristics. The propensity score analysis was conducted to evaluate changes in the risk of distant disease presentation during the pandemic by estimating the adjusted percentage of patients diagnosed with metastatic disease in 2020 and 2021 compared to the pre-COVID-19 period. AI/AN: American Indian/Alaska Native. PI: Pacific Islander.

## Data Availability

The original data presented in the study are openly available in the SEER database.

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
