# Peer review of "Disruptions in Lung Cancer Detection During COVID-19"

_cancers, 2024, doi:10.3390/cancers16234001_

Round 1

Reviewer 1 Report

Comments and Suggestions for Authors

The current paper reviews the impact of COVID 19 on the incidence of lung cancer detection. COVID 19 has led to an increase in lung cancer detection in late, symptomatic stages and banned elective healthcare for a limited but long enough period of time.
The authors used the SEER database for a period of 20 years, but they included in the study only the patients with microscopic lung cancer diagnosed between 2018-2021. Please clarify the extent of the original time frame (2001-2021) compared with the real time frame of the study (2018-2021).
The inclusion and exclusion criteria are clearly presented, but please clarify the time frame difference here.
The statistical methods are appropriate, and the results are clearly presented.
The study is valuable because it is a real-world demonstration of the impact of the pandemic on lung cancer screening. In some middle- and low-income countries where screening is deficient, lung cancer screening has increased compared with the pre-pandemic period, but this study highlights an important aspect. In populations with limited access to health care, low income, marginalized patients, the screening due to symptomatic COVID 19, lead to the discovery of lung cancer in advanced stages. This may be superimposable with the limited statistics in middle and low income countries.
Although the realistic limitations of the study may not cover the true magnitude of the findings, the current paper is valuable and may serve as a benchmark pilot study for non-U.S. middle and low income countries.
After answering my question, I propose to accept the paper in its current form.

Reviewer 2 Report

Comments and Suggestions for Authors

Thank you for the opportunity to review this manuscript. This paper may contribute to the lung cancer prevention and pandemic preparedness. However, some of the points are not very clear to me and need further explanation.

  1. Materials and methods: “We used both the SEER-17 and SEER-22 databases. These are maintained by the National Cancer Institute (NCI), which does not monitor the accuracy of the data reported, the statistical analyses performed, or the conclusions drawn by the authors.” Please describe more about the SEER-17 and SEER-22 databases and also provide the url links of the databases.
  2. “Patient cohort”. Need to provide more information related to the sample selection. It is not clear how the data was selected or filtered, whether there were any missing data involved or treated. The description of the inclusion and exclusion criteria here is for the actual diagnosis of lung cancer. Does that mean in the dataset, there is not a clear diagnosis of “lung cancer” but you need to look for the information on “ICD code” etc? Given the readers who are not familiar with the SEER-17 and SEER-22, it is confusing.
  3. It would be necessary to reorganize the results part. The results were reported with big amount of numbers and had little connections between results. It would be beneficial to only report significant and meaningful results.
  4. Reading this manuscript makes me really confused. The data used are lung cancer incidences. But the authors' emphasized point is lung cancer detection. The incidence rate of lung cancer is related to the diagnosis of lung cancer. However, the decreased diagnosis of lung cancer is not only attributed to the decreased detection/screening of lung cancer, but also may due to large amount of people who have lung problems died before they are diagnosed, screened or treated in the pandemic. Statements in the paper are confusing when the authors used detection and incidence/diagnosis interchangeably, e.g., “Importantly, by 2021, we documented partial recovery in lung cancer detection” should this be detection or diagnosis or incidence?
  5. Discussion: “These findings align with prior studies that reported an increased likelihood of distant disease at diagnosis during the pandemic, particularly in its early stages[27],[28].” “Distant disease … in its early stage” what does this mean?
  6. “… we observed decreased odds of SCLC diagnosis in the first year of the pandemic” “One potential explanation for this unexpected result is the incidental detection of early-stage lung cancers through imaging obtained for suspected COVID-19 infection.” Need to explain more to connect the reason to the result.
  7. “However, the persistently low detection rates for advanced-stage disease emphasize the need for targeted interventions to ensure all populations benefit from these recovery efforts.” What is the rationale here for this statement? Need more explanation.
Comments on the Quality of English Language The English language is used appropriately in the paper. However, the logic and connection/flow between sentences are not easy to understand.

Reviewer 3 Report

Comments and Suggestions for Authors

This study uses SEER database to demonstrate that: The COVID-19 pandemic caused a significant drop in lung cancer detection, especially in rural areas, non-Hispanic Black and Asian individuals, and women. While detection rates improved in 2021, disparities remained, highlighting the need for targeted interventions to reach underserved populations.

Introduction

Please try to brief introduction about the LDCT screening effect with equality issue such as gender difference, self-paid model or other socioeconomic problems.

References:

Evaluating Efficiency and Adherence in Asian Lung Cancer Screening: Comparing Self-paid and Clinical Study Approaches in Taiwan

YJ Wu, EK Tang, FZ Wu

Academic Radiology

Method

Please address the study flowchart to illustrate the study design.

Result and discussion

Try to use visual abstract to illustrate the COVID effect on lung cancer detection rate.

Please discuss more about the study limitation about no questionnaire study about the people’ attitude about lung cancer screening during the COVID period.

Reviewer 4 Report

Comments and Suggestions for Authors

The paper is very well written; I only have some minor suggestions to improve its quality.

1. Page 2 of 12, Line 92: "We used both SEER-17 and SEER-22." The authors may want to clarify what SEER-17 and SEER-22 represent—whether the numbers refer to the years of data collection, release, or something else. A brief explanation could help readers, especially those unfamiliar with SEER, understand the dataset better.

2. Page 3 of 12, Lines 116–117: "We then utilized a previously described method." The authors may consider adding some details on the previously described method, such as how the expected numbers of lung cancer incidence were calculated.

3. Page 3 of 12, Line 132: "We included interaction terms between the year of diagnosis and each covariate." It would be helpful if the authors could list these covariates.

4. Discussion Section: The discussion is very well written. If possible, the authors could add insights into the global public health implications, such as whether similar undiagnosed cases and disruptions in lung cancer detection occurred worldwide during COVID-19.
